# NemoTrainer: Automated Conditioning for Stimulus-Directed Navigation and Decision Making in Free-Swimming Zebrafish

**DOI:** 10.3390/ani13010116

**Published:** 2022-12-28

**Authors:** Bishen J. Singh, Luciano Zu, Jacqueline Summers, Saman Asdjodi, Eric Glasgow, Jagmeet S. Kanwal

**Affiliations:** 1Department of Neurology, Georgetown University Medical Center, Washington, DC 20057-1460, USA; 2Department of Oncology, Georgetown University Medical Center, Washington, DC 20057-1460, USA

**Keywords:** fish, conditioning, software, auditory discrimination, hearing, learning, attention, spatial working memory, reward, vision

## Abstract

**Simple Summary:**

We report here the development of a new apparatus and method for the automated control of animal training and discrimination learning using a microcontroller and custom software. We describe our methodology and deploy the system to train and test learning and decision making in freely swimming adult zebrafish. This animal model is of increasing interest among scientists from many different fields, including ethology, ecology, bioinformatics, and neurology. As a proof of concept, we obtain data on audiovisual discrimination learning. Our system of training can be scaled down or up in size to train small invertebrate species or larger vertebrate, including mammalian species.

**Abstract:**

Current methods for associative conditioning in animals involve human intervention that is labor intensive, stressful to animals, and introduces experimenter bias in the data. Here, we describe a simple apparatus and a flexible, microcontroller-based conditioning paradigm that minimizes human intervention. Our methodology exploits directed movement towards a target that depends on spatial working memory, including processing of sensory inputs, motivational drive, and attentional mechanisms. Within a stimulus-driven conditioning paradigm designed to train zebrafish, we present a localized pulse of light via LEDs and/or sounds via an underwater transducer. A webcam placed below a glass tank records fish-swimming behavior. For classical conditioning, animals simply associate a sound or light with an unconditioned stimulus, such as a small food reward presented at a fixed location, and swim towards that location to obtain a few grains of food dispensed automatically via a sensor-triggered, stepper motor. During operant conditioning, a fish must first approach a proximity sensor at a remote location and then swim to the reward location. For both types of conditioning, a timing-gated interrupt activates stepper motors via custom software embedded within a microcontroller (Arduino). “Ardulink”, a Java facility, implements Arduino-computer communication protocols. In this way, a Java-based user interface running on a host computer can provide full experimental control. Alternatively, a similar level of control is achieved via an Arduino script communicating with an event-driven application controller running on the host computer. Either approach can enable precise, multi-day scheduling of training, including timing, location, and intensity of stimulus parameters; and the feeder. Learning can be tracked by monitoring turning, location, response times, and directional swimming of individual fish. This facilitates the comparison of performance within and across a cohort of animals. Our scheduling and control software and apparatus (“NemoTrainer”) can be used to study multiple aspects of species–specific behaviors as well as the effects on them of various interventions.

## 1. Introduction

Behavioral assays are commonly used to measure various types of learning. Previously available assays incorporate apparatuses such as shuttle-boxes [1,2,3,4,5], forced choice T-mazes [6,7], three compartment mazes [8,9] for studying the effects of nicotine [10,11], and plus-mazes [12] to conduct behavioral research in different species. Most of these assays require some level of human intervention, which can make the experiments cumbersome, time-consuming and sometimes costly. Most importantly, animal handling can stress the animals and introduce noise in the data [13,14]. Conditioned place preference assays [15,16] are simple and fast, but typically require fear conditioning, which is itself stressful to the animal. Here, we describe a new type of behavioral assay that minimizes stress and provides a fast, reliable way to train and test an animal’s ability to discriminate between sensory stimuli. Most importantly, our assay provides the possibility to test not just learning but also navigational dynamics associated with a stimulus-directed task. This together with modern animal-tracking options offers an enriched analysis of learning and attentional dynamics for studying defects in cognition, motivation, and movement patterns, including decision making. The apparatus and software described here can be used for training animals using either classical or operant conditioning (see also [17]), and/or for studying their decision-making behavior under semi-natural, free-swimming conditions. Learning can be tracked using the timing of angular (turning angle) and translational (swim trajectories) movements as well as the posture at a time-stamped location of each individual within the testing environment. This facilitates comparison of performance within and across a cohort of animals.

Our training method and behavioral assay uses reward-based, stimulus-triggered spatial trajectories in freely moving animals. In the case of fish, this translates into directional swimming activity. Decisions to make appropriate turns from memory are important in this assay, but unlike a T-maze, it does not involve a forced choice among two alternatives at a fixed location and time point. Rather, the fish are free to swim in any direction and reach their goal naturally by choosing any sequence of multiple turns and swim trajectories. The apparatus is fully controlled by computer software communicating with a microcontroller that can be easily programmed through a graphical, user-friendly interface or otherwise. The custom software generates event-synchronized, short (10 to 30 s) video data files that can be analyzed using automated imaging software developed by others [18]. The training and testing apparatus together with scheduling and control software, dubbed “NemoTrainer” (vers. 1.0), was developed specifically to communicate with an external microprocessor (Arduino UNO r3; www.arduino.cc, accessed on 20 December 2022). One way we implemented NemoTrainer was through Java (https://www.oracle.com/java/, accessed on 20 December 2022) running on a personal computer (PC), to allow high-resolution video recording and cloud storage capabilities that are well beyond that of either an Arduino or a Raspberry Pi microcontroller. An Arduino provides a convenient interface to integrate the analog functionality of a robotic setup. Here, we tested learning and memory in free-swimming adult zebrafish as an application of NemoTrainer.

The zebrafish (*Danio rerio*) is a widely used model organism in the fields of genetics, oncology, developmental biology [19], and neurobiology [20,21,22]. Most recently, research has focused on a genes-to-circuits-to-cures approach [23,24,25]. Growing evidence suggests that zebrafish can play an important role in elucidating the genetic and neural mechanisms underlying multiple neurological disorders, such as [26,27] Parkinson’s [24] and depression. This motivated us to develop an inexpensive and efficient method for the automated training of individual as well as small cohorts of zebrafish. Since zebrafish exhibit many social behaviors, analogous to those present in humans [28,29,30], they are also excellent animal models for understanding the neural and genetic contributions to the onset of autism spectrum disorders [31]. Furthermore, targeted gene mutations in combination with behavioral studies in this organism can contribute to our basic understanding of particular neural circuits involved in learning, recall, and decision-making.

We have fine-tuned and tested the hardware and software, and describe behavioral trials to demonstrate associative conditioning in adult zebrafish. The data suggest that NemoTrainer can be used to test visual and auditory discrimination in zebrafish. Below, we provide details of the hardware and software and discuss its usefulness and limitations as well as the care needed to successfully conduct experiments using NemoTrainer.

## 2. Materials and Methods

### 2.1. Design Philosophy

Our goal here was to create a user-friendly, low-cost, automated behavioral training system that could be easily replicated and expanded for high throughput use with multiple animals. As such, we designed a space-efficient system, and employed readily available, cost-effective technology and materials.

Using our system, it is possible to perform both auditory and visual training by means of a fully programmable, schedule-based combination of proximity sensors, LEDs, and an output transducer. Using either a unique food delivery setup in combination with aversive stimulation, reward- and fear-based conditioning, respectively, can be implemented to entrain and test fish behavior related to learning and decision making. Here, we focus on an exclusively reward-based paradigm. To present a comprehensive package for data collection and analysis, we also briefly report our findings on video analysis and animal tracking software available at the time of this study.

### 2.2. Training Apparatus

Animals used in this study were housed and maintained in the zebrafish core facility at Georgetown University. All animal experiments carried out here were formally approved by the Georgetown University Animal Care and Use Committee and conform to the National Institute of Health Guide for the Care and Use of Laboratory Animals. Experiments were conducted in a quiet, light-controlled (14:10 light–dark cycle; ambient light 14 Lux) corner room to minimize disruption by external auditory and visual cues. Experimental animals were transported from the core zebrafish facility at Georgetown University and placed in an acclimation tank in the experimental room. Animals were restricted from being fed for 3 days (normal feeding interval) immediately preceding a learning trial so that they were motivated to explore and seek a food reward. This also minimized any stress from handling and transportation between the routine housing and the experimental room. Zebrafish were trained in a circular, 30 cm diameter × 9 cm tall, glass tank. An artificial plant was placed at the center of the tank to simulate a somewhat naturalistic environment. To minimize reflections, eliminate unwanted visual cues, and create uniform and diffuse lighting, the top of the experimental tank was covered with a translucent diffuser sheet and the inside tank wall was lined with an adhesive opaque plastic strip. This arrangement also improved fish vs. background contrast for video recording with a webcam (Logitech HD Pro 920, Vienna, VA, USA) placed ~30” below the tank to eliminate occlusion of the image by any above-tank objects, such as the feeder apparatus (Figure 1). To provide an in-tank associative directional cue, one-inch wide, different colored (red or green) plastic strips were mounted on opposite sides of the tank behind each feeder port. The training tank also contained two digital, infra-red proximity sensors (Sharp GP2Y0D810Z0F; Pololu Inc., https://www.pololu.com/product/1134, accessed on 20 December 2022), a thermometer probe, and an aerator (a complete parts list is included in the Appendix B). The sensor was set up in an always-on configuration by placing a reflector in front so that a beam break by the fish sent a positive trigger pulse from its voltage-out pin to the Arduino [32]. The sensor itself was mounted either above the water surface or outside the glass tank. The latter placement prevents water vapor from corroding the circuitry on the sensor board. An LED on the sensor board indicated its always-on condition for monitoring its stability and reliability. The LED’s reflection in the water allowed for video monitoring of its status remotely and also provided feedback to the animal when it triggered the sensor. For presenting sound cues, an underwater transducer (model M-10, Lubell Labs, Inc., https://www.lubell.com/, accessed on 20 December 2022) was placed in the center of the tank. A sound level meter (model 407735, Extech Instruments, Inc., https://www.testequipmentdepot.com/extech/soundmeters/407735.htm, accessed on 20 December 2022) positioned at the water surface indicated a sound pressure level of ~65 dB SPL, but is expected to be much higher (~100 dB SPL) in the underwater sound field to be within the hearing sensitivity of adult zebrafish [33].

Feeding ports located on opposite sides of the circular tank (180 degrees apart) consisted of funnels mounted above the tank where the colored plastic strips were affixed. A stepper motor (SOYO, https://www.pololu.com/product/1205, accessed on 20 December 2022), designated as the “feeder motor”, activated a rapid 4 to 5 degree, clockwise/counterclockwise rocking motion (enhanced feed) or unidirectional circular motion (simple feed) of a plastic tube with a small hole (“feeder”) to drop food particles through each funnel into the water.

The feeder tube was attached to the shaft of a stepper motor mounted on half-inch PVC tubing. To enable reward presentation (food drop) at different locations, the feeder motor was mounted to a robotic, L-shaped arm, which could be rotated by a second stepper motor, (SOYO, https://www.pololu.com/product/1476, accessed on 20 December 2022), designated as the “arm motor” as shown in Figure 1. This motor moved the arm to a user-specified angular location measured in degrees from an arbitrary starting point. This two-motor assembly was positioned above the tank (Figure 1). The feeder dropped the food after a delay (measured in seconds from the start of the recording) for classical conditioning. For successive feeds at different locations, the arm motor moved at the desired speed to the location specified in degrees after a user-specified delay (see Appendix A). For operant conditioning, the fish triggered the proximity sensor to obtain food at a predetermined location.

### 2.3. Hardware and Software Design Features

We used Arduino, a dynamic microcontroller, for controlling external devices such as stepper motors, servos, sensors, LEDs, etc. that are ordinarily not configurable by a standalone computer. The Arduino can be used to independently program up to three LEDs and one sound channel to present visual and/or auditory stimuli at any time after trial onset. The Arduino, though capable of controlling external devices, is not designed to control programs installed on a PC. For the purposes of monitoring behavior and presenting visual and/or auditory stimuli at specified time points, our training system required automated, synchronous control of PC-based software, namely, Debut (https://www.nchsoftware.com/capture/index.html, accessed on 20 December 2022) and VLC (https://www.videolan.org/, accessed on 20 December 2022).

One way to use Nemotrainer is to employ a custom program written in Java that operates within an Arduino-integrated software development environment. For this, Ardulink (vers. 0.4.2), an open-source Java library (https://github.com/Ardulink/, accessed on 20 December 2022), was used to facilitate real-time control of Arduino UNO r3 (or LEONARDO, www.arduino.cc, accessed on 20 December 2022) microcontrollers. Ardulink features a collection of Java Swing (https://www.javatpoint.com/java-swing, accessed on 20 December 2022) components pre-configured to communicate with Arduino over USB, allowing the rapid development of a graphical user interface (GUI). Together, Arduino and Ardulink controlled all connected devices, including both stepper motors (SOYO; https://www.pololu.com, accessed on 20 December 2022) as well as auditory and visual stimulus generators.

Another way to effectively use NemoTrainier is to employ an Arduino script (https://github.com/SinghB13/NemoTrainer, URL accessed on 20 December 2022) in combination with GoBetwino software (http://mikmo.dk/gobetwino.html, accessed on 15 January 2019)—an event-driven application controller running on the PC to synchronize timing of Arduino-connected device functions with actions by PC-based software. Functions defining arm movements, feeder movements, and power to LEDs, were written according to the type of training to be conducted. Gobetwino-specific commands defining PC-based keystrokes were incorporated into functions to control PC-based software, such as Debut for video capture, thereby allowing specific recording and sensory stimulus presentation durations.

The software component of NemoTrainer sets up delay-based triggers to plan and execute scheduled events (see flow chart in Figure 2). Events are organized as a series of “trials”, “runs”, and “reps”. Additional runs can be planned using the same or modified parameters by specifying the time and day for a sequence of runs. If operant conditioning is selected in the interface, the time window during which a sensor is triggered to elicit a food reward is time-limited to the duration of a stimulus. If the fish does not trigger the sensor during this time window, presentation of the stimulus repetition is terminated and, after a user-specified delay, is followed by the next rep for that run, or alternatively, the next “run”. If classical conditioning is selected, food reward is given after a fixed time delay for a specified number of runs. Our experiments typically consisted of 4- to 5-day trials of 6–8 daily runs of up to 6 reps each. NemoTrainer, however, has the capability to automatically execute scheduled runs for many weeks without any user intervention. Together, a set of runs comprises a trial.

Runs begin by starting a video recording. The start time of a recording is used as a reference point for subsequent delay-based actions. The timing for stimulus presentations is user-specified by means of a delay (in seconds) from recording onset via the user-defined scheduler interface (Figure 3). This way, LEDs and auditory stimuli can be presented independently and/or simultaneously. As an event continues, depending on the type of behavioral conditioning chosen, either sensor triggers or additional delays are employed to precisely plan actions such as time of food reward, arm re-positioning, and the next “rep”. Additional user-specifiable interface settings include record-end delay, LED selection and duration, auditory stimulus selection (by specifying the location of a saved sound file on the PC) and duration, degrees of arm rotation, and simple or enhanced feed. Reps conclude with the end of a video recording, the length of which is based on the sum of all delays and triggers plus an additional delay to extend the recording if desired. Once the number of reps specified per run is reached, the run is concluded and the software waits until the time of the next scheduled run to begin the same.

### 2.4. Computer Interface and Operation

The user-defined scheduling interface (Figure 3A) was designed for manual entry of all run and trial control parameters prior to starting an experimental trial. This information sets up the design of the experiment by setting up time delays for various devices, video recordings, etc. to be activated over several days. Each scheduled run begins with onset of video recording for a defined time interval.

Launching NemoTrainer software package from the PC opens a window with four dropdown menus at the top and the user-defined scheduling interface in the space below (Figure 3A). Details of each dropdown menu are as follows:

*File*: contains options to save a trial, load an existing trial, and quit Nemo Trainer.

*Ardulink*: contains an option to establish or terminate a link between Nemo Trainer, and Arduino, and associated hardware (motors, sensors, LEDs). This link must be established prior to beginning a trial to enable communication between software and Arduino-connected hardware.

*Tools*: contains options to specify pin settings for LEDs and test individual actions such as motor movements, LED output, sound output, and video recording. Sound files should be placed in one folder, which should be specified in the tools menu.

*Help*: contains license and contact information for the creators of Nemo Trainer.

The scheduler-interface consists of 19 columns for all of the variables for which the parameter values can be defined. These values are entered via the “New Action/Timer Dialog” button located at the bottom of the screen. Clicking this button launches the action/timer dialog through which run parameters are configured (details are provided in the next section). Clicking on the adjacent button labelled as the “Open Training Dialog” launches a window containing the finalized trial configuration. At the bottom right of this dialog box are a “24H Remix” check-box and 4 buttons. Two of these buttons provide the option to start and stop a trial. The 24H mix feature repeats the entire set of runs every 24 h. The “Random mix” button provides the option to change the order of the runs listed in this dialog box and can be used to randomize the presentation of different sounds. Both options are only available when runs are scheduled using 24 h time as opposed to day and date. Otherwise, the Training Dialog window is identical to the table created interactively within the scheduler-interface below the menus in the opening window. Each row in the scheduler-interface (Figure 3A) is created by clicking on the “Action/Timer Dialog” button and filling in the relevant parameters in the form fields either directly or via dropdown menu options. This interface enables any user to comprehensively plan and automatically begin multiple trials.

Below the scheduler-interface space are two buttons on the left to configure trial parameters and two buttons on the right to turn sensors on and off. In addition, two drop-down selectors above the sensor buttons are used to specify sensor pin connections. The two sensors can be connected to any of the numbered pins on an Arduino, but the numbers selected here should match the physical connections. These sensor activity monitors enable quick and simultaneous access to live feedback from each of the two sensors. The pins specified here also specifies within the NemoTrainer software which pins to monitor during a trial. Clicking each button enables live feedback on the status of the sensors, which can be viewed regardless of whether a trial is ongoing or not. During online feedback, when a sensor is triggered, the circle above the button corresponding to that sensor becomes bright and text changes to “High” to indicate that it was triggered. Conversely, when a sensor is not triggered, the circle above the corresponding button remains dark and the text on its side changes to “Low”.

Finally, at the bottom far-right corner of the interface panel is an icon indicating the status (connected/disconnected) of the link between NemoTrainer and Arduino. An Ardulink sketch must be uploaded to the Arduino prior to launching NemoTrainer to establish the link. Since the Arduino communicates with the PC via a USB port, the custom program must establish a connection to the USB port, and for the duration of a trial, maintain an uninterrupted line of communication with the Arduino. This enables live sensor and event feedback without any additional hardware normally required for feedback. Furthermore, it allows users to pause, make changes to parameters, and resume trials without having to end and start a new trial.

Due to the precision needed for arm placement, food delivery, and all other training parameters, it is critical to ensure that all functions work as expected prior to starting a trial. As such, we have developed a testing window within NemoTrainer through which users can independently test LEDs, sound presentation, video recording software, and motor movement. In addition, users can specify the parameters of a run and conduct a test run, which in addition to the previously mentioned functions, also tests delays and sequence of events, to ensure that all values specified for a run meet their requirements.

Launching the Action/Timer Dialog opens a window with six boxes (Figure 3B). A description of these boxes for execution of NemoTrainer software is listed in Appendix B. Altogether, up to two sensors and three LEDs can be addressed using drop-down menus. The direction and speed of motor movement can also be user-determined for the duration of a trial.

### 2.5. Example Training Paradigm

Zebrafish can demonstrate visual learning in as few as 4 training sessions [8]. In one of the training paradigms, we conducted four days of training consisting of six training sessions per day. Given the well-established visual acuity in zebrafish [34], we paired different colored LEDs with different sound stimuli when testing their discrimination ability. Three distinct sounds were synthesized using software available at Wavtones.com (https://www.wavtones.com/functiongenerator.php, accessed on 20 December 2022), and filtered and normalized (RMS) using Audacity (https://www.audacityteam.org/, accessed on 20 December 2022). Sounds 1 and 2 were paired with specific food delivery locations, and sound 3 was played without food delivery to discourage nonspecific association of sound with food delivery. On day 1, sounds 1 and 2 were presented in an alternating order within consecutive runs for operant-controlled food delivery. During any repetition of a run, if a fish triggered the correct sensor during sound playback, a small food reward was delivered automatically within a second. On day 2, in addition to alternating sounds, two seconds after the onset of sound presentation, an LED, either green or red, was illuminated where food was delivered. The LED, a visual stimulus, provided the directional cue and followed the onset of the auditory target cue to promote paired associative learning. Zebrafish are highly dependent on their visual system to hunt for food [35] so a visual stimulus can facilitate training of other sensory modalities. Each run served as an opportunity for fish to associate an auditory or visual cue with a food reward. Runs were scheduled to take place at 8 AM, 10 AM, and 12 PM, followed by three more runs at 4 PM, 6 PM, and 8 PM after a break to match their daily feeding/activity rhythm in their natural habitat. Within each run, a user-specified number of repetitions (usually 4 to 6) of the same control parameters were specified. LEDs were not illuminated during any runs on day 5 when testing auditory discrimination learning.

### 2.6. Measures and Analysis

Videos from every repetition were automatically recorded at 30 frames/s (fps) using Debut. Upon onset of recordings, we schedule 10 s as a delay from recording onset to observe baseline behavior. After this, sounds & LEDs were presented and fish were given the opportunity to earn food reward. Recordings terminated 5 s after food delivery, resulting in videos approximately 30 s long. Several tracking methods were tested. Three of the available programs, Ctrax (https://ctrax.sourceforge.net/, accessed on 20 December 2022), wrMTrck (https://www.phage.dk/plugins/wrmtrck.html, accessed on 20 December 2022), a plugin for Image-J (https://imagej.nih.gov/ij/download.html, accessed on 20 December 2022) to track fish movements and record frame-by-frame positional coordinates, and idTracker (https://www.idtracker.es/, accessed on 20 December 2022), were all able to track multiple (4 to 8) zebrafish with different levels of success. Videos were analyzed automatically using idTracker [18]. After opening idTracker, a video file is selected for tracking. Several parameters are adjusted to ensure proper data collection, e.g., the number of fish in the video to be tracked must be entered in the “Number of individuals” box. The “Number of reference frames” must be less than the number of frames that are being tracked. These settings can be saved and used later or the “Start” button can be used to begin tracking. Once tracking is complete, a figure of the fish tracks appears and files containing tracking data are saved in the folder containing the tracked video. It is recommended that each video be placed in its own individual folder before it is opened in idTracker. Lighting is a critical factor in the successful application of automated tracking software. For videos with less-than-optimal lighting, one can resort to manual tracking. This provided verifiable quantitative data on fish position. Image-J allows for automated as well as manual tracking of multiple zebrafish. Although this method is slow and involves manual marking of fish in each video frame, it offers a reliable choice for any critical short-duration (<10 s) videos. Additional information on use of software packages is included in Appendix C.

Behavior was recorded prior to and post-sound and LED onset to compare arousal using swim-speed and darts. To measure learning during classical conditioning, three zones were specified with increasing distance from the feeder location. We selected two time-windows, a window prior to and another post-sound/LED onset for analysis. During these two time-windows, fish proximity to food delivery locations was calculated based on either manual or software extracted coordinates of fish locations within individual frames. This was also a robust measure for operant conditioning, although the goal was to have the fish consistently trigger the sensor rather than simply be in the proximity of the sensor during the 8 s (typically 4 s for classical conditioning) time-window after sound playback.

## 3. Results

We successfully performed automated training and testing of zebrafish as well as recorded and analyzed their social and swimming behavior using available animal tracking software. We describe both exemplary and quantitative data indicating that zebrafish can be conditioned to respond to audiovisual stimuli via classical and operant conditioning. To test our setup and training paradigm, we trained fourteen zebrafish (seven for classical, and seven for operant conditioning) over a relatively short time period (three to four days). Generally, from the first feeding on day 1 of training and onwards, zebrafish showed a significant increase in arousal following sound onset. This was reflected in the increase in the frequency of turns and changes in swim speed.

### 3.1. Classical Conditioning

Swim trajectories for seven fish during a classical conditioning trial are shown in Figure 4. Almost all individuals swimming randomly in different locations and directions within the tank (Figure 4A) oriented themselves with different latencies towards the correct direction following the onset of LED/sound cues (Figure 4B). Correct direction is defined as the direction corresponding to the side of the tank where the food was delivered when a specific sound was played through the underwater speaker. This matched the placement (side) of the target LED. To physically define and constrain the boundaries of the target vs. non-target side for each run, we placed Plexiglass dividers with entry/exit holes offset from the center to prevent rapid swimming towards and away from the target side, which the fish tend to do when aroused. Though this helped to momentarily capture the fish at their target swim side, it also prevented some individuals from reaching the LED target and feeding location, which they could see through the Plexiglass, because in their haste to reach the target by a direct swim trajectory, they failed to swim through the entry hole offset to one side and instead kept pushing against the Plexiglass divider. Learning was indicated by a significant decrease in distance from the target side within 4 s or less from the onset of the audiovisual stimulus compared to before stimulus onset (Figure 4C). We used multiple repetitions of short (200 ms duration) bursts of either pure tones or frequency-modulated (FM) sweeps centered at 1 kHz, well within the frequency range of hearing in zebrafish [33,36,37,38].

For classical conditioning, sham training included random presentation of sounds and/or LEDs as the control condition during pre-training. Sham trial runs showed no significant difference overall in either orientation or distance from the vertical plane at which the divider was located. These data were tested for 1 s and 3 s pre- and post-stimulus presentation in two different cohorts (one-way ANOVA; *p* = 0.793; n = 6 and one-way ANOVA; *p* = 0.719; n = 6). Shams were sounds which were not associated with the reward and were presented randomly to make the reward association nonspecific to the sound to be tested. Immediately following sound and LED onset, fish tended to follow a preferred swim pattern. Sometimes this involved swimming to the center of the tank first, probably for orientation purposes, and then to the zone of their choice, which took approximately two seconds. Therefore, the time window for analysis after sound and LED onset was selected accordingly. Widening the analysis time-window or shifting it to begin after 3 s from sound onset incorporated behavioral noise so that significant learning was no longer detectable in the data. This partly resulted from rapid swimming back to the opposite side of the tank, thereby confounding the results.

During some classical conditioning trials with multiple fish in the tank, we noticed zebrafish chasing each other around in a way resembling dominant and submissive interactions [39], in which a submissive individual swims away from a dominant, more aggressive, conspecific. Frequently, rather than moving towards the reward-side of the tank following the onset of an audiovisual cue, dominant individuals began chasing conspecifics, delaying their presence within the food-reward zone past the time-window assigned for analysis. For these reasons, we further developed our software to train individual fish within an operant conditioning paradigm.

### 3.2. Operant Conditioning

For operant conditioning, we presented food once at each of the feeding locations to allow fish to orient themselves to the training environment immediately after their introduction to the test tank. As explained in the methods section, in this paradigm, a fish freely triggered a sensor to elicit a reward, which was accompanied by presentation of audiovisual stimuli. Most fish failed to respond successfully to sound presentation alone in the early runs, but their attempts at triggering the sensor as well as success rates increased within a 10 to 12 h time window (Figure 5) and peaked within a 24 to 48 h time frame. More data are needed, however, to claim that zebrafish, and for that matter any fish species, can discriminate complex sound features, such as the direction of FMs. This is a focus of some of our ongoing work.

Examples of three separate “runs” recorded at the beginning, during and at the end of training illustrate navigational learning in zebrafish (Figure 5). Swim direction and trajectories were visualized using manual data entry from Image-J, from individual frames of a 30 frames/s video recording. In the “untrained” condition (Figure 5A), the animal continues to swim away from the sensor and around the tank and sensors. In the “partially trained” condition, the animal responds to the sound and moves to the right location to trigger the sensor, but fails in its attempt (Figure 5B), and in the “trained” condition, it succeeds in triggering the sensor twice: first, swimming in front of it (red arrows) to trigger a food reward, and then circling back (red arrows) to trigger it once again before proceeding to the feeding location (Figure 5C). In this setup, the location of the food drop was fixed in between the two sensor locations.

### 3.3. Learning Dynamics

We measured learning success by scoring a “1” for a successful trigger and a “0” for a failure to trigger. We obtained a learning curve by plotting a smoothed kernel fit of the density of 0s and 1s at successive time intervals. The main plot in Figure 6 shows successes and failures for a single trial over time. The inset shows the results of a statistical analysis using repeated measures ANOVA to test main effects of test subject (fish) and “cumulative time” for learning. Between-subject learning rates were highly variable, yet success at triggering the sensor (stimulus-driven directed swimming) was significantly higher (*p* < 0.05) over time (rejection of null hypothesis for no differences). Significance (*p* < 0.05; n = 7) was typically reached after 10 test runs. A linear regression model yielded a good fit (R^2^ = 0.85) for actual vs. predicted responses. In summary, our results demonstrate the effectiveness of NemoTrainer. Our results demonstrate the potential to test for sound discrimination in zebrafish via spatial navigation within either a conditioned or operant learning task. In our study, zebrafish learned to associate sound stimuli with LEDs and visual markers placed at opposite sides of the tank. They used available visual stimuli (air stone location, different colored markers fixed at sensor locations, etc.) to orient themselves towards the sensor and feeding location with a relatively high degree of accuracy as determined by the general orientation of head direction. NemoTrainer enabled rapid spatial orientation and swimming in reward-conditioned zebrafish. Whether adult zebrafish can respond selectively and reliably to different types of sound stimuli needs further investigation.

## 4. Discussion

Here, we developed and used NemoTrainer to study navigational decision making in adult zebrafish in response to audiovisual stimuli. The user-friendly graphical interface that we created for this purpose, however, may be applicable to various setups designed for behavioral training in other species. We show that, using NemoTrainer, zebrafish can be trained to respond to and/or discriminate between LEDs and auditory cues including pure tones and frequency-modulated sounds, and potentially other parameters of complex, natural sounds, such as frog croaks and catfish barks [40,41]. However, at this juncture, we do not claim that our findings contribute to new scientific knowledge; this will require trials with additional animals to address various contingencies and test additional acoustic parameters. Rather, we present our results as a proof of concept of NemoTrainer. NemoTrainer minimizes human interference and experimenter bias, and at the same time frees the experimenter to invest their time in monitoring multiple setups, data analysis, and other tasks. It can provide a fast and efficient means for behavioral screening of neurologic drugs and to potentially test the effects of CRISPR-based as well as optogenetic modification of neural circuits.

The merit of fish models in genetics, neuroscience, pharmacology, and toxicology is now a matter of record (e.g., see [19]). The next and ongoing push is to extend the zebrafish model to pursue questions of behavioral neuroscience, especially genetic and drug manipulation of behavior, an undertaking that requires valid, reliable, and efficient methods of behavioral training and assessment [3,42,43,44,45,46]. In addition to their potential for clinical applications, automated training paradigms facilitate the examination of general principles of goal-driven behavior, attentional mechanisms, decision making within solitary and social conditions, sensory discrimination capabilities, as well as neural mechanisms of learning and memory.

### 4.1. Notable Program Features

The scheduling interface is an important and convenient feature of our Java-based training software. It facilitates a high level of customization and allows one to set up and visualize the training paradigm for as many days as required. Our training of animals is incentive- rather than fear-based. Fear-conditioning can lead to stress and deviations from natural behavior. In contrast to the classical conditioning methods, operant conditioning builds on each instance of observation/response and reinforces an overall goal that the animal must learn to accomplish.

Testing for associative learning typically entails the presentation of the conditioned stimulus without a reward at the end of training. Each such test can lead to extinction effects, which can confound the results over a series of tests, i.e., slowly counteract whatever learning took place previously. Our protocol avoids potential extinction effects because test runs can be set up (see scheduler interface in Figure 3A) to alternate with training runs at preset intervals. This is possible because the availability of recorded data off-line allows the experimenter to track learning throughout the trial and make planned interventions to maximize learning efficiency via rapid assessment of learning parameters, such as head-turns, relative position within tank, and distance from target. The ability to determine optimal training and experiment duration is important because sequence learning efficiency diminishes if the duration of training exceeds an optimal level [47]. Furthermore, with appetitive conditioning, satiation can also contribute to this effect as suggested by the periodic humps in the learning curve in Figure 6.

Since training and testing during stimulus-directed swimming involves sensory discrimination, memory, recall, and navigational elements, any of these can be targeted for testing drug effects. Moreover, the apparatus can be used with single individuals as well as small groups of four to eight zebrafish during classical conditioning, allowing one to study the effect on learning in transgenic models of autism spectrum disorders [48,49]. Zebrafish models for Alzheimer’s and Parkinson’s are also well-developed and tested primarily at the cellular and molecular levels [24,26,27]. Our simple-to-use training paradigm adds the possibility of testing drug treatment efficacy on behavior.

### 4.2. Audiovisual Learning in Zebrafish

Zebrafish are very good at spatial memory tasks as shown by the simple method of delivering food at alternate locations within a tank. In an earlier report [3], a spatial memory task was triggered by a tapping cue after which the fish had to choose the correct side (relative to the placement of a red card) of their tank to receive food. In our paradigm, fish swam directly to a specific location of the tank. Incorporation of a narrow angular zone has the potential to have the fish choose between multiple directions to swim towards, to test an array of auditory or visual stimuli. This can involve the fish making a series of sequential decisions that cavefish are known to be capable of making [50]. In the paradigm adopted for the present study, the choice depended on the side a colored LED was turned on and a sound presented. Zebrafish are highly responsive to LEDs given their well-developed visual system [34].

The training and testing method adopted here can also be used to advance our basic understanding of the behavior and neurobiology of any sensory and motor system in any fish species. The hearing ability of fish, in general, is well-established and considered to be important for their survival in the aquatic environment [41,51,52,53]. However, their ability to discriminate between complex sounds has not been systematically tested at either the behavioral or neurophysiological levels [40,54]. A recent study employing brain-wide imaging showed the differential activation of neurons within the larval zebrafish brain to a variety of sounds [54]. NemoTrainer has the potential to be used to further examine this ability in almost any species. Furthermore, since fish, unlike mammals, are able to regenerate hair cells within their sacculus [55], a sound stimulus-based behavioral assay can also be used to track regeneration and re-establishment of function after hair cell ablation and drug treatment studies without sacrificing animals. Similarly, visual discrimination can be rapidly tested by using multiple, different-colored LEDs along the circumference of the tank and rewarding the fish only when they swim to an LED of a particular color at a specific location in the tank.

### 4.3. Imaging and Tracking Options

Excellent packages have been designed to track small freely moving organisms, such as fruitflies and round worms. idTracker was designed to specifically track zebrafish, although it has been tested on fruit flies, ants, and mice as well [18]. Using various program features, idTracker can be used to identify territoriality and leadership attributes of an individual, making it a particularly useful tool in studying social behavior. One of the major challenges with tracking zebrafish is accurately identifying individuals once they swim past or over one another. idTracker resolves this issue by determining and assigning a specific fingerprint to each individual fish and using that fingerprint to identify the individual throughout the tracking process. The fingerprint of each fish is used in all frames of the recording, allowing idTracker to track with a 99.7% accuracy even when fish cross or overlap. By identifying each individual fish, idTracker greatly reduces errors in accurately tracking fish, both when they are isolated or crossing one another frequently. Furthermore, fingerprints of individual fish can be saved and used later in different videos to allow for progressive data collection from multiple video files.

idTracker uses contiguous pixel value parameters to identify and track zebrafish. From our usage of idTracker, we found that homogenous lighting and background conditions greatly improved tracking results whereas a greater contrast between the fish and their background allowed for more effective segmentation and tracking. Therefore, when adjusting the lighting for viewing and video-recording our observation arena, we spent significant time and effort in optimizing the lighting conditions. Most recently, DeepLabCut [56], a machine learning algorithm, has shown a lot of promise. This together with idTracker.ai [57] may prove to be the methods of choice for tracking zebrafish. DeepLabCut does require manual training and the availability of PCs equipped with a GPU. Neither of these software packages nor the appropriate hardware was available to us at the time of this study.

### 4.4. Caveats and Future Directions

Any behavioral assay requires careful monitoring and patience as all animals may neither behave the same as others nor learn at the same rate, and sometimes they do not learn at all. A few precautions noted here will ensure the smooth running of each assay with a new animal. We have observed that learning proceeds more rapidly if zebrafish are oriented to the setup by dropping food once or twice at the appropriate locations. The number of food particles dropped per cycle is fairly reliable if sifted for uniformity in size. The actual amount of food dropped depends on various parameters, such as the speed of rotations, number of turns, size of hole in feeder tube, etc. Regardless, some random variation in the amount of food dropped should not affect training. Food should be checked every other day and refreshed as needed to avoid clumping due to high humidity just above the water surface, but this period could be extended by using a small heater to create a gentle flow of dry air at the level of the feeder tube.

It is also important that training starts as soon as fish are introduced into the tank and are still engaged in active swimming to explore their environment. During this active swimming period, it is critical for fish to learn that swimming in certain locations, i.e., near sensors, can result in stimulus and/or reward presentation. This, in effect, establishes that the environment is dynamic rather than static and thus leads to further exploratory behavior during a trial. Otherwise, if a fish has exhaustively explored its environment without any stimulus or reward presentation during this critical exploration period, its behavior becomes more sedate and subsequently leads to decreased responsiveness during a trial. NemoTrainer can be further modified to incorporate this feature.

Troubleshooting is an important part of a custom hardware–software setup. Therefore, we note here some of the issues we encountered during our experimentation with this setup. When using a microcontroller with pin connections, it is important to periodically check that connections have not become loose or exhibit increased resistance because of oxidization due to a high humidity environment. To make the system amenable to student handling, we adapted phone jacks for the easy connection and disconnection of motors and sensors from the Arduino input and output terminals. The digital proximity sensors are used as motion sensors in our setup. When placed above water for generating a vertical beam, their exposed contacts should be coated with a thin layer of vinyl for protection from moisture. While sensors can get power from the Arduino, they are more stable if they are connected to a separate 5 V power supply. It is important to check that the stepper motors get adequate power through the driver. Each driver has a small setscrew, which can be turned clockwise or anti-clockwise to adjust the power supplied to the stepper motor. Too little power can result in jerky motion and too much power can make the motors run hot.

Due to the automated and modular nature of our system, it should be suitable for use in experimentation with other types of small fish commonly used in laboratories, including goldfish and catfish that have been extensively used in behavioral and neuroscientific studies. Furthermore, the physical training apparatus can be easily expanded or reduced in size by using a longer arm and a larger tank to house and train more fish simultaneously. The larger motor chosen for this application is powerful enough to hold up to 4 kg/cm and is therefore capable of handling additional weight by a longer, heavier arm.

In summary, we have developed and demonstrated a fully automated method for the training and testing of freely swimming zebrafish within a stimulus-dependent navigational task that may require recall of either allocentric (sensory association) or egocentric (directional swimming) cues. NemoTrainer can be used to address many behaviorally, neurologically, and genetically relevant questions in an effective and efficient manner in an appropriate species.

## 5. Conclusions

Our results suggest that our training apparatus and software (NemoTrainer) can be used for either classical or operant conditioning of freely moving animals. This can allow experimenters to test sensory discrimination capabilities of any species as well as allow high-throughput screening of drugs to study their effect on learning and memory, or for any other type of behavioral assessment. Here, we used NemoTrainer to test the ability of zebrafish to discriminate between different colored LEDs and sound patterns. We demonstrated that adult zebrafish can learn to respond to food reward in a navigational memory task in <5 days and sometimes in <3 days without any human intervention.

## 6. Patents

Two patents have been granted from the development of the methodology reported and tested here. These are listed below:Systems and Methods for Automated Control of Animal Training and Discrimination Learning; US Patent No.: 10,568,305 B2; Date of Patent: 25 February 2020. Assignee: Georgetown University.Systems and Methods for Automated Control of Animal Training and Discrimination Learning; US Patent No.: 11,369,093 B2; Date of Patent: 28 June 2022. Assignee: Georgetown University.

## Figures and Tables

**Figure 1 animals-13-00116-f001:**
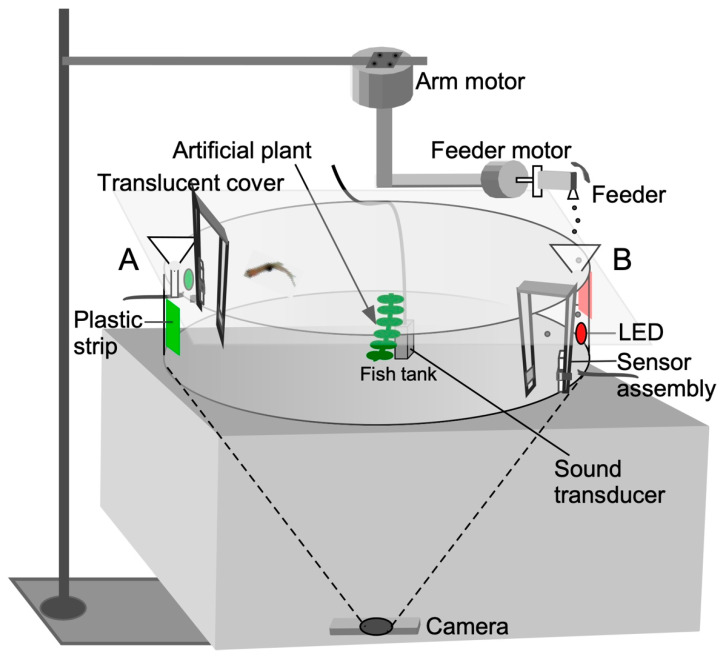
Diagrammatic representation of the audio-visual training apparatus for operant conditioning. For classical conditioning, a sensor trigger is not required. Fish-training/testing trials on stimulus-directed swimming were conducted in groups for classical conditioning and individually for operant conditioning. See text for additional details of the setup and Figure 2 for software control.

**Figure 2 animals-13-00116-f002:**
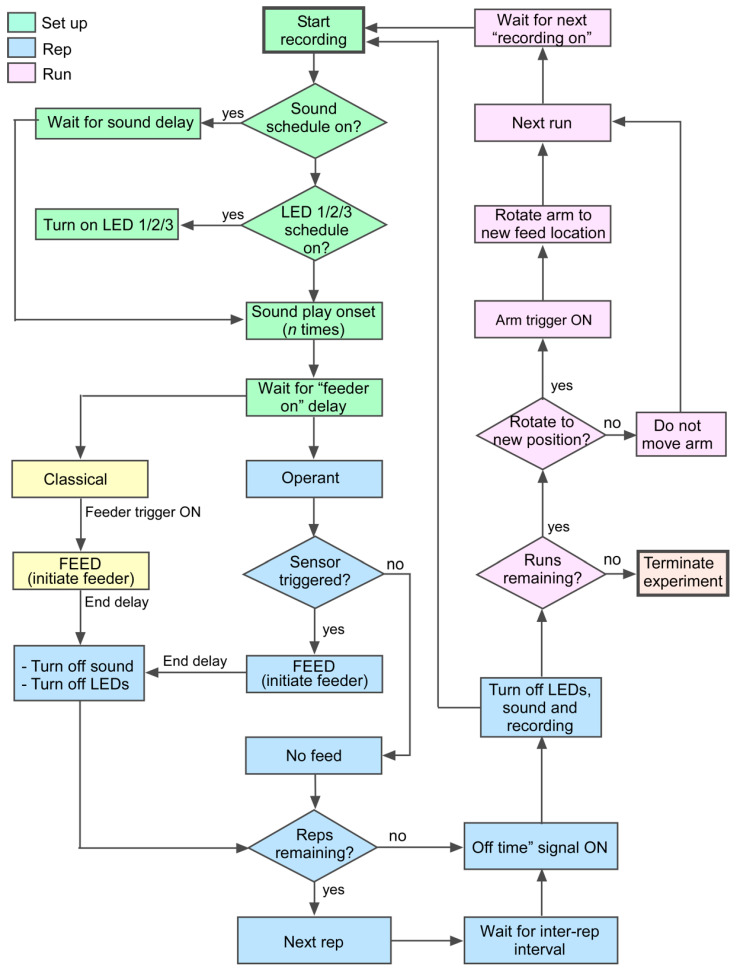
Flowchart depicting the algorithm for automated conditioning. The training procedure assigns user-defined delays for turning “on” and “off” light and sound or any other type of stimulus as part of the setup (green). Stimulus repetition (blue) provides multiple opportunities in close succession within each of multiple daily runs (pink) for the animal to learn the task.

**Figure 3 animals-13-00116-f003:**
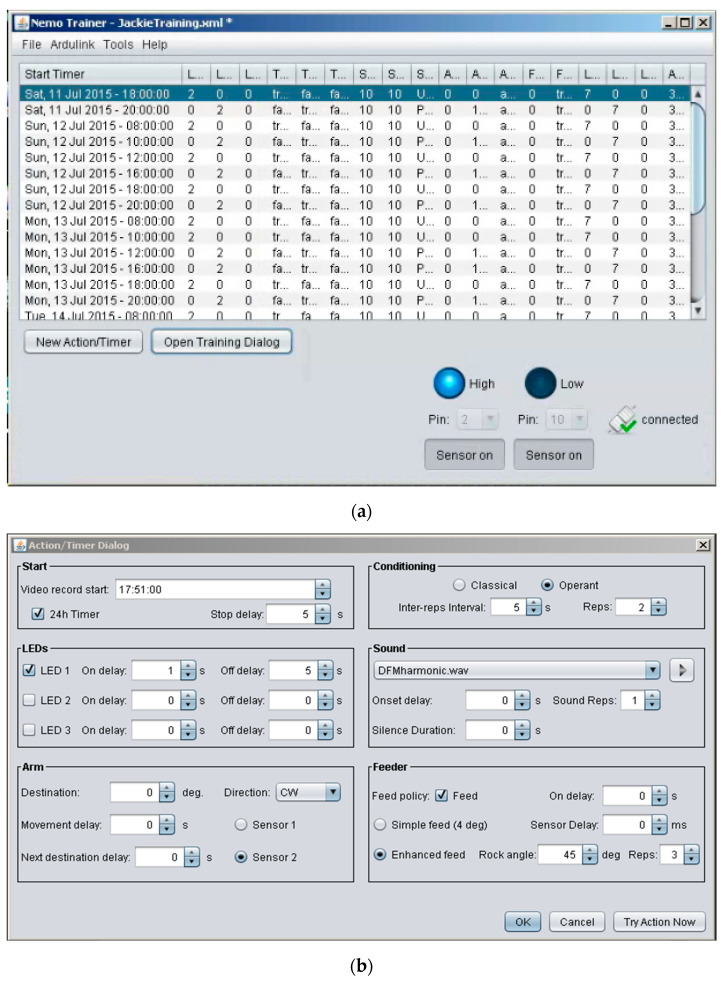
Screen captures of the user interface for the Java facility. (**a**) Scheduler-interface showing the training schedule for communication with Arduino. (**b**) Computer interface for monitoring sensors and control of associative conditioning. User-definable settings enable either classical or operant conditioning via customized multi-day scheduling and precise control of stimulus parameters for training.

**Figure 4 animals-13-00116-f004:**
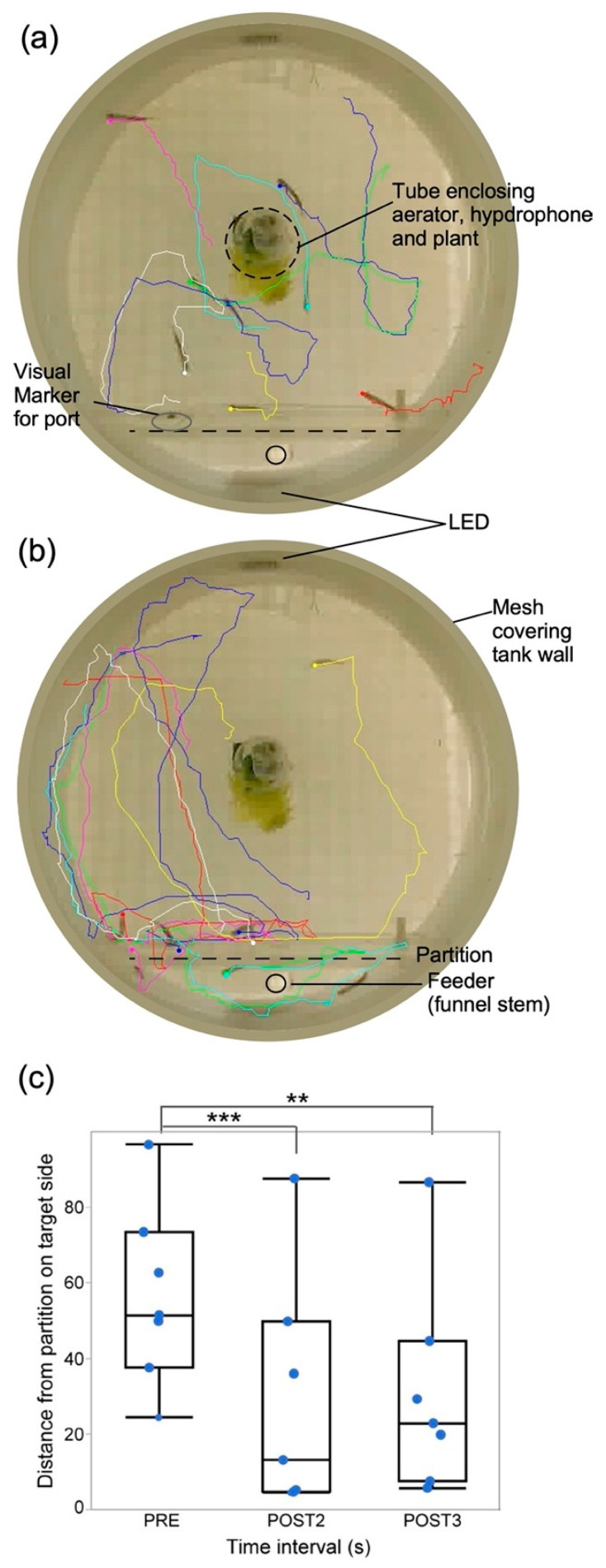
Tracking individual fish behavior. Tracks of swim paths of seven fish created using idTracker from video recordings (**a**) before, and (**b**) during presentation of a short, 4 s duration sound stimulus. Tracks begin at locations 1 s pre- and terminate 4 s post-sound onset. (**c**). Box plots and jittered scattergrams showing distances of fish (n = 6) from correct (target) side after training in response to the presentation of sounds within a single trial (4). Data were obtained from 1 s before (PRE) and during the second (POST2) and third (POST3) second post-stimulus. Shorter distances from target indicate better learning. On average, fish were closer to target during POST2 and started to wander away during POST3. ** indicate statistically significant and *** indicates highly significant difference between means (horizontal line within each box). One stationary fish trapped behind the divider was not tracked and another starting at the divider was not included in the analysis. Aerator and underwater speaker are on the right and feeder tube on the left. Perimeter shows glass wall reflection as the image was captured by a camera placed below the tank.

**Figure 5 animals-13-00116-f005:**
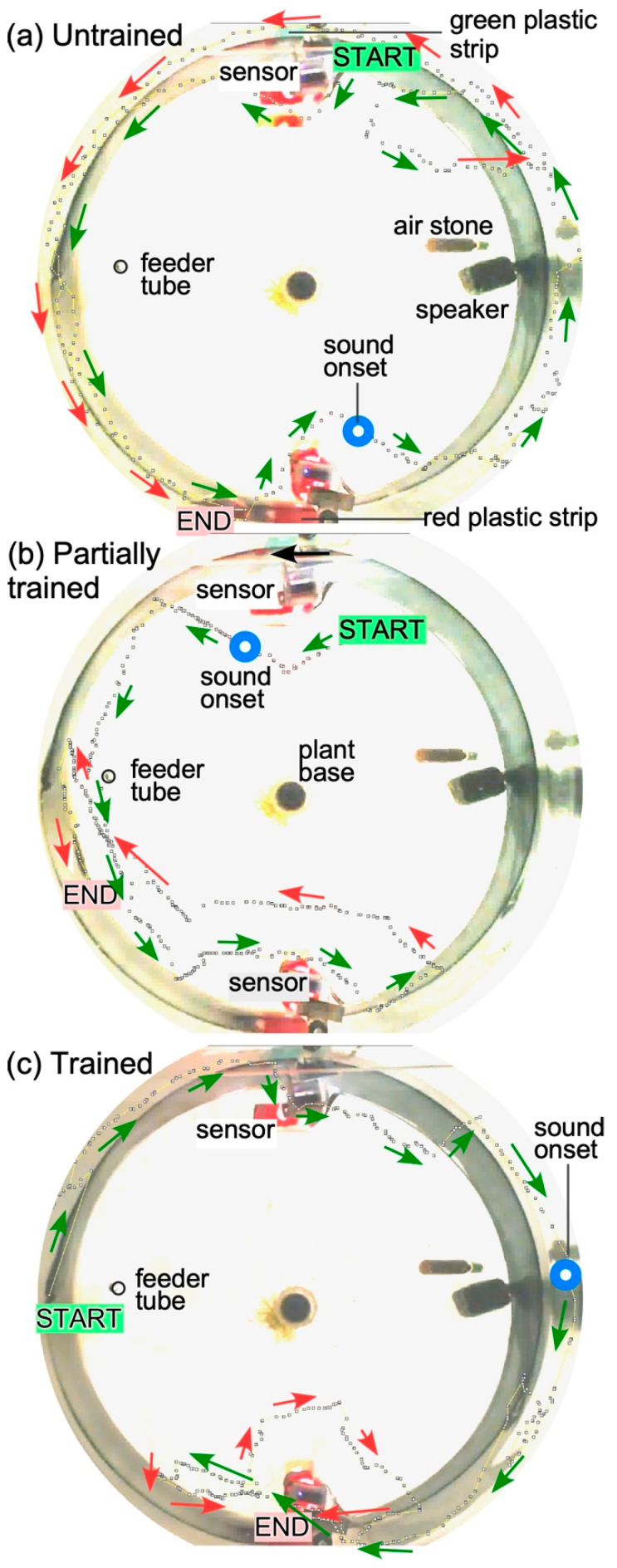
Progression of learning in a single zebrafish during operant conditioning. (**a**): Fish does not show any overt response to the presentation of a sound stimulus. (**b**). Fish shows awareness and attempts to trigger the sensor on the correct (red) side, but is unsuccessful. (**c**). Fish successfully triggers the sensor twice to receive a food reward. Dotted lines indicate the swimming path obtained from individual frames in the recorded video of the fish. A shortening of the distance between successive dots along the swim path indicates slowing down of the fish after hearing the sound. Blue circles indicate time point of sound onset superimposed on the fish’s swimming path. Arrows next to the trajectory indicate the heading of the fish. Green arrows indicate trajectory early in time and red arrows indicate trajectory later in time. Sensor activation/beam-disruption was indicated by a small LED flash on the sensor board that is reflected back in water (red rectangles in the image).

**Figure 6 animals-13-00116-f006:**
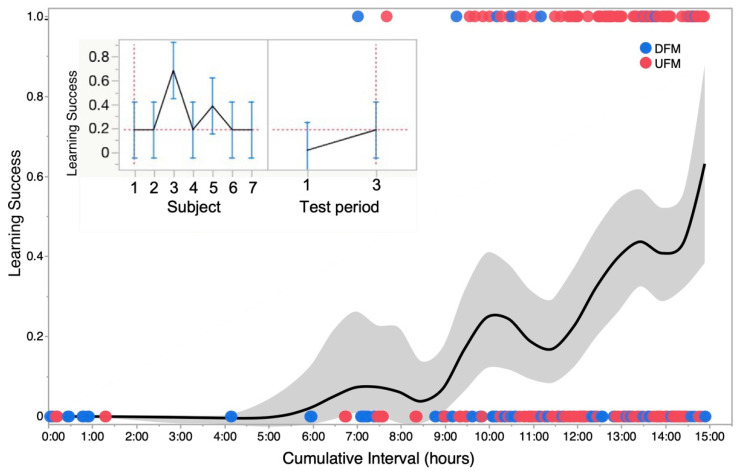
Learning curve and variability in operant conditioning across animals tested. The main plot shows a smoothed kernel spline fit to show the learning trend for a binomial distribution of failure (0) and success (1) in successive trials for one zebrafish summed for two types of sound stimuli—upward and downward frequency modulations labeled as UFM and DFM, respectively. Grey zone shows standard deviation of data points from mean plot at each time step or run. Inset: Profile plots generated from a repeated-measures analysis of variance (ANOVA) showing the cumulative success (mean ± stand. deviation) of seven zebrafish tested. Panel on left shows that overall, there was a significant (*p* < 0.05) “subject” effect mainly because of one individual (animal #3). Panel on right shows that significant (*p* < 0.05) learning was observed across all seven animals based on success rates for triggering sensor within the 8 s time window when averaged for the first and third set of runs occurring within a 12 to 24 h time frame (time interval over which learning occurred varied with individual fish).

## Data Availability

The software is publicly available for download at following URL: https://github.com/SinghB13/NemoTrainer.

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
