# Peer review of "NemoTrainer: Automated Conditioning for Stimulus-Directed Navigation and Decision Making in Free-Swimming Zebrafish"

_animals, 2022, doi:10.3390/ani13010116_

Round 1
Reviewer 1 Report
Title: "NemoTrainer: Apparatus and Software for Conditioning of 2
Stimulus-directed Navigation and Decision Making in Freely 3
Behaving Animals"
In this work the authors presented a localized pulse of light via LED’s and/or sounds via an underwater transducer.
A webcam placed below a glass tank recorded fish swimming behavior. For classical conditioning, animals simply 28
associated a sound or light to an unconditioned stimulus, such as a localized food reward, and swim 29
towards that location. During operant conditioning, a fish must interrupted an infrared beam at one 30
location to obtained a small food reward at the same or different location. For both types conditioning, 31
a timing-gated interrupt activated robotic-arm and feeder stepper motors via custom software 32
controlling a microprocessor (Arduino). “Ardulink”, a JAVA facility, implements Arduino- 33
computer communication protocols. In this way, full automation of stimulus-conditioned 34
directional swimming was achieved. Precise multi-day scheduling of training, including timing, 35
location and intensity of stimulus parameters, and feeder control was accomplished via a user-friendly 36
interface. Learning was tracked by monitoring, turning, location, response times and directional 37
swimming of individual fish. This procedure facilitated the comparison of performance within and across a cohort 38
of animals. The authors claim that they demonstrated the ability to train and test zebrafish using visual and auditory stimuli. 39
The authors also claimed that their scheduling and control software and apparatus (NemoTrainer) is effective to screen neurologic 40
drugs and test the effects of CRISPR-based and optogenetic modification of neural circuits on 41
sensation, locomotion, learning, memory and attention
General comment: The work should be reworked according to the following issues:
2.4.1. The Action/Timer Dialog
*) This paragraph should be reworked and the list of items should be avoided.
Figure 5. Tracking a single fish in three runs during operant conditioning. A: Fish does not show 475
any overt response to the presentation of a sound stimulus. B. Fish shows awareness and attempts 476
to trigger the sensor on the correct side, but is unsuccessful. C. Fish successfully triggers the sensor 477
twice to receive a food reward and swims to the feeder location to retrieve the reward. Lines 478
interrupted by small squares track movement of fish. Red filled bold circles indicate time points in 479
the fish’s swimming path when the sound was played. The dots in the tracks indicate the position 480
of the fish within individual frames in the recorded video. Arrows next to the trajectory indicate the 481
heading of the fish. A shortening of the distance between successive squares along the swim 482
trajectory indicate slowing down of the fish after hearing the sound. Proximity sensors generated
*) This figure as well as the figure caption are not totally clear, please rework.
Figure 6. Learning curves and individual variation. A smoothed kernel spline fit to show the general 536
learning trend for a binomial distribution of failures (0) and success (1) of trials for one zebrafish. 537
Grey zone shows standard deviation from mean plot. Inset: Profile plots generated from a repeated- 538
measures analysis of variance (ANOVA) showing the success rate (mean ± stand. deviation) of 7 539
zebrafish tested within NemoTrainer. Panel on left shows that overall, there was a significant (P < 540
0.05) “subject” effect mainly because of one individual (“A3”). Panel on right shows that significant 541
(P < 0.05) learning was observed based on success rates for triggering sensor within the 8 s time 542
window when averaged for the first and third set of tests occurring within a 12 to 24-hour time 543
frame (time interval over which learning occurred varied with individual fish).
*) See the previous comment. Please rework the figure caption to improve its clarity. Better labels within the figure could be beneficial.
Author Response
We greatly appreciate this reviewer's effort and feedback on our manuscript. Regarding the checked-off comment on English language, it was not clear to us which particular stylistic changes are needed. However, we have carefully read through the entire manuscript, and made changes in a number of places to improve the language, clarity and flow. These are indicated in the tracked-changes version of the revised manuscript.
More specifically, we have reworked the section on "Action/timer dialogue box". We have deleted all of the list of items and moved them into the Appendix. Any remaining text was moved to the previous section. This helps with the flow of reading.
We have also improved the clarity of figure legends and figures 5 and 6, as suggested by this reviewer.
Once again, we thank the reviewer for their feedback that prompted us to improve clarity of writing and hence the overall impact of the manuscript.
Reviewer 2 Report
In the current interesting manuscript the authors describe a new a user-friendly, low-cost automated behavioral training system, the NemoTrainer that minimizes human interference and experimenter bias. This training and testing method used in zebrafish except for the advantage to identify individuals by using a fingerprint, it can be used to advance basic understanding of the behavior and neurobiology of any sensory and motor system in any fish species. The manuscript is well written, the introduction explains the aim if the study, the method is described in details and the discussion successfully comments on the data. The authors themselves note the need for further experiments and they also mention possible technical problems to pay attention.
Comments.
Title:The title is very general. The words ‘preliminary results’ and ‘zebrafish’ should be included in the title
Methods: According to the authors, 12 zebrafish were used. Please specify at each experiment the number of zebrafish used (i.e in fig 6 7 zebrafish were used)
Author Response
We greatly appreciate this reviewer's encouragement, interest and recognition of the value of our work. We have further improved the clarity and accuracy of the text, and also address specific reviewer comments below.
As suggested by this reviewer, we have modified the title to include the word "zebrafish" in the title. Regarding inclusion of "preliminary results", we clarify that this is primarily a methods paper, and so we do not stress either in the title or in the abstract, results that may advance our scientific knowledge of the sensory and other capabilities per se of zebrafish. We have excluded use of "preliminary" in most places in the manuscript to avoid giving a wrong impression of our methodological advance. In addition, we now include the following statement in the discussion section:
"We show that using NemoTrainer, zebrafish can be trained to respond to and/or discriminate between LEDs and auditory cues, including frequency-modulated sounds, and potentially other parameters of complex, natural sounds, such as frog croaks and catfish barks [35,36]. However, at this juncture we do not claim that our findings contribute to new scientific knowledge; this will require trials with additional animals to address various contingencies and test additional acoustic parameters. Rather, we present our results as proof-of-concept of NemoTrainer."
We thanks, reviewer for the question on the number of animals tested. We have now corrected and clarify this in the manuscript. We successfully tested 14 animals and indicate the numbers used (7 for classical- and 7 for operant conditioning). Many more animals were used at various stages in the development phase of our methodology, but are not included here.